# Network Pharmacology Study to Elucidate the Key Targets of Underlying Antihistamines against COVID-19

Ki-Kwang Oh [ID], Md. Adnan [ID] and Dong-Ha Cho *

Department of Bio-Health Convergence, College of Biomedical Science, Kangwon National University, Chuncheon 24341, Korea; nivirna07@kangwon.ac.kr (K.-K.O.); mdadnan1991.pharma@gmail.com (M.A.)
* Correspondence: chodh@kangwon.ac.kr; Tel.: +82-33-250-6475

**Abstract:** Antihistamines have potent efficacy to alleviate COVID-19 (Coronavirus disease 2019) symptoms such as anti-inflammation and as a pain reliever. However, the pharmacological mechanism(s), key target(s), and drug(s) are not documented well against COVID-19. Thus, we investigated to decipher the most significant components and how its research methodology was utilized by network pharmacology. The list of 32 common antihistamines on the market were retrieved via drug browsing databases. The targets associated with the selected antihistamines and the targets that responded to COVID-19 infection were identified by the Similarity Ensemble Approach (SEA), SwissTargetPrediction (STP), and PubChem, respectively. We described bubble charts, the Pathways-Targets-Antihistamines (PTA) network, and the protein–protein interaction (PPI) network on the RPackage via STRING database. Furthermore, we utilized the AutoDock Tools software to perform molecular docking tests (MDT) on the key targets and drugs to evaluate the network pharmacological perspective. The final 15 targets were identified as core targets, indicating that Neuroactive ligand–receptor interaction might be the hub-signaling pathway of antihistamines on COVID-19 via bubble chart. The PTA network was constructed by the RPackage, which identified 7 pathways, 11 targets, and 30 drugs. In addition, GRIN2B, a key target, was identified via topological analysis of the PPI network. Finally, we observed that the GRIN2B-Loratidine complex was the most stable docking score with −7.3 kcal/mol through molecular docking test. Our results showed that Loratadine might exert as an antagonist on GRIN2B via the neuroactive ligand–receptor interaction pathway. To sum up, we elucidated the most potential antihistamine, a key target, and a key pharmacological pathway as alleviating components against COVID-19, supporting scientific evidence for further research.

**Keywords:** antihistamines; COVID-19; neuroactive ligand–receptor interaction; GRIN2B; loratadine

## 1. Introduction

An initial outburst of inflaming lung was reported firstly at Wuhan in Hubei Province, China, and its severity was alerted to the World Health Organization (WHO) by the Wuhan Municipal Health Commission on 31 December 2019 [1]. Currently, the COVID-19 pandemic is continuing to outbreak around the world, with everyone in the world exposed to enormous risk of infection from COVID-19 [2,3]. More significantly, although most pathological symptoms are identical to pneumonia, nevertheless, the majority of COVID-19 patients do not have any signs, thus have been contagious to others as 'silent spreaders' [4]. Due to the disproportionate supply of COVID-19 vaccines, around 25% of the world's population may not have access to the COVID-19 vaccines until at least 2022 [5]. The situation may be getting worse with the COVID-19 pandemic, particularly, causing tremendous pain in low- and middle-income countries [6]. Hence, some clinicians administer antiviral drugs and antihistamines as a considerable therapeutic strategy for COVID-19 patients [7,8]. A recent study suggested that the utilization of antihistamines is safe for COVID-19 treatment without any adverse effects [9]. Furthermore, histamine receptor antagonists dampen long-COVID symptoms such as inflammatory pneumonia, and especially Loratadine of

the H1 receptor antagonists was reported to inactivate SARS-CoV-2 effectively [10,11]. Antihistamines are prescribed to relieve a wide range of inflammatory responses due to their potent anti-cytokine production [12]. For instance, some reports demonstrated that antihistamines are related to alleviating allergic rhinitis, allergic conjunctivitis, sinusitis, bronchitis, peptic ulcer, acid reflux, and gastritis, all of which are associated directly with the overproduction of cytokines [13]. Most antihistamines are known as antagonists of the H1 receptor and/or the H2 receptor involved in the production of histamine [14]. Most notably, Loratadine of the cycloheptene-piperidines species could interfere with SARS-CoV-2 infection, which is a new generation H1 receptor antagonist with highly selective properties, as well as less central adverse effects [10]. The severity of lung inflammation has been related to histamine secretion after SARS-CoV-2 infection [15]. Moreover, research shows that the H1 receptor blockade could negatively affect the production of interferon gamma and might be effective in dampening inflammation driven by the SARS-CoV-2 infection [16]. Although antihistamines may be effective therapeutic candidates against COVID-19, the lack of scientific evidence has limited application to COVID-19 patients. Thus, we aimed to uncover the most promising antihistamine(s), target(s), and pharmacological pathway(s) against COVID-19 via network pharmacology strategy. Network pharmacology can decipher the pharmacological mechanism(s) of drugs from a holistic viewpoint, which represents a paradigm shift from "one target, one drug" to "multiple targets, multiple drugs" [17,18]. Besides, network pharmacology is an effective systemic approach to guide in drug repurposing [19]. Drug repurposing for COVID-19 treatment on orchestrated network pharmacology can also accelerate the discovery of significant drugs against COVID-19 [20]. In this research, network pharmacology was utilized to investigate the most significant potential antihistamine(s) against COVID-19. Firstly, antihistamine drugs (32 drugs) were selected via drug browsing websites. The selected antihistamine drugs and COVID-19 related targets were identified by public bioinformatics and cheminformatics. Secondly, final overlapping targets were analyzed to explore potential pharmacological pathway of antihistamines on COVID-19. Thirdly, PPI networks on topological analysis were constructed through the RPackage. Finally, MDT was performed utilizing AutoDock Tools-1.5.6 to evaluate the best conformer between potential targets and antihistamines. The workflow is exhibited in Figure 1.

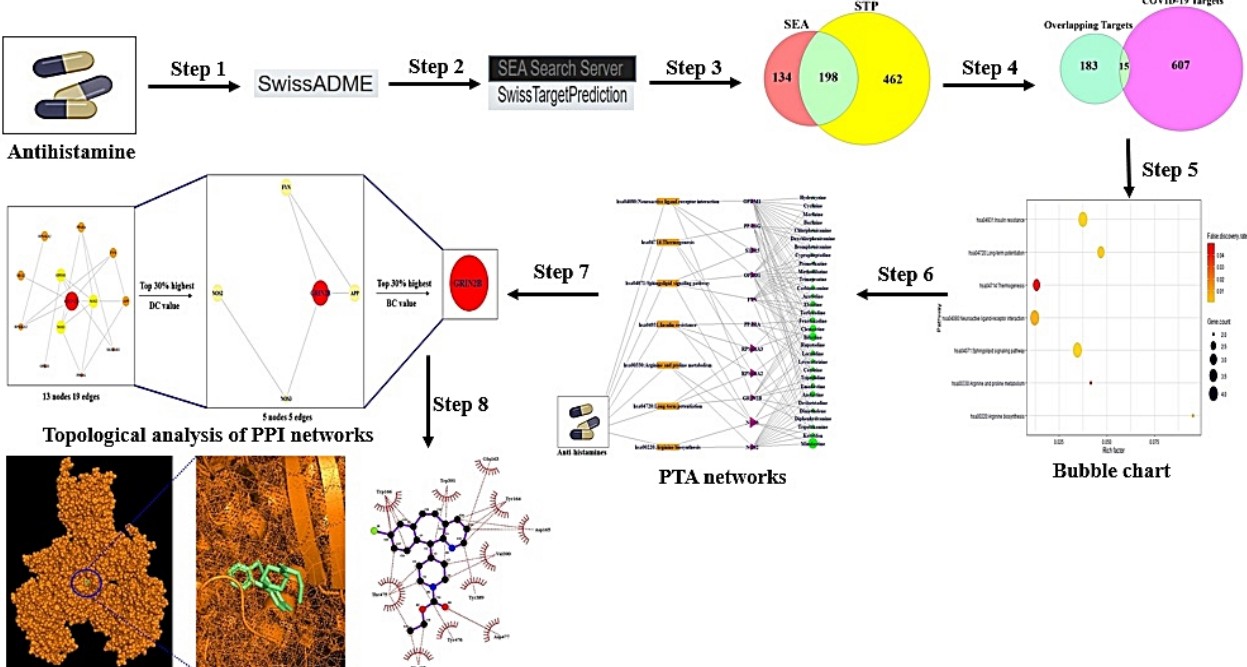

**Figure 1.** Workflow of network pharmacology analysis of antihistamines against COVID-19.

## 2. Materials and Methods

### 2.1. Antihistamines Connected to Targets or COVID-19 Related Targets

Antihistamines prescribed on the market were utilized in this study. The targets of antihistamines were identified via Similarity Ensemble Approach (SEA) (https://sea.bk slab.org/) (accessed on 22 December 2021) [21] and SwissTargetPrediction (STP) (http://www.swisstargetprediction.ch/) (accessed on 22 December 2021) [22] with "Homo Sapiens" mode. Besides, targets associated with COVID-19 were identified by selecting COVID-19 in PubChem (https://pubchem.ncbi.nlm.nih.gov/) (accessed on 22 December 2021). Additionally, the Topological Polar Surface Area (TPSA), extracted by SwissADME (http://www.swissadme.ch/index.php) (accessed on 22 December 2021), is indicated to confirm the antihistamines' cell membrane permeability, significantly, the threshold of TPSA value is less than 140 Å to adopt as orally administered drugs [23]. The final overlapping targets between antihistamines and COVID-19 targets were visualized by Venn Diagram Plotter.

### 2.2. Pharmacological Pathway Enrichment Analysis on a Bubble Chart

A bubble chart was plotted by the RPackage based on STRING database (https://string-db.org/) (accessed on 23 December 2021). There was an observational KEGG (Kyoto Encyclopedia of Genes and Genomes) pathway indicating a key mechanism (lowest rich factor) between antihistamines and COVID-19 related targets.

### 2.3. Pathways-Targets-Antihistamines (PTA) Network Construction

The Pathways-Targets-Antihistamines (PTA) network was constructed by the RPackage on STRING database (https://string-db.org/) (accessed on 26 December 2021). In the PTA network, different colors and shapes (nodes) stand for the signaling pathway (orange square), target (red triangle), and antihistamine (green circle). Gray lines (edges) showed the relationships of PTA. The PTA network was used to construct a size map according to degree of values. In the PTA network, the size of red triangles represents the number of relationships with pathways; the size of green circles stands for the number of relationships with targets. The combined networks were constructed by the RPackage.

### 2.4. Topological Analysis of Protein-Protein Interaction (PPI) Network

The PPI network analysis was performed using the RPackage on STRING database (https://string-db.org/) (accessed on 26 December 2021). We performed this to identify crucial targets in the topological analysis from PPI network. Firstly, targets with the greatest Degree of Centrality (DC) values in the upper 30% were sorted for subnetwork assembly using the RPackage. Next, target(s) with the greatest Betweenness Centrality (BC) values in the top 30% from subnetwork were considered as core target(s).

### 2.5. Preparation for MDT of Antihistamines

The ligands were converted from .sdf format from PubChem into .pdb format utilizing Pymol, subsequently, the ligands were converted into .pdbqt format via AutoDockTools-1.5.6.

### 2.6. Preparation for MDT of Targets

A key target of a hub pharmacological pathway, i.e., GRIN 2B (PDB ID: 7EU8), was identified by RCSB PDB (https://www.rcsb.org/) (accessed on 27 December 2021). The target selected as .pdb format was converted into .pdbqt format via AutoDockTools-1.5.6.

### 2.7. MDT of Antihistamines—A Key Target

The ligands were docked with a key target using AutoDock4 by setting up 4 energy ranges and 8 exhaustiveness as default to obtain 10 different poses of ligand molecules [24]. The active site's grid box size was x = 128.688 Å, y = 128.088 Å, z = 133.365 Å. The 2D binding conformer was identified through LigPlot +v2.2 2 (https://www.ebi.ac.uk/thornton-srv/software/LigPlus/) (accessed on 27 December 2021). [25]. We identified the

best scoring conformer (the lowest binding energy), which was considered as the most significant antihistamine to ameliorate the symptoms of COVID-19.

## 3. Results

### 3.1. Physicochemical Properties of Antihistamines

A total of 32 antihistamines were selected through drug browsing websites and some literature. The antihistamines' physicochemical properties and TPSA (Topological Polar Surface Area) are displayed in Table 1. The chemical structures of these antihistamines are exhibited in Figure 2.

**Table 1.** Physicochemical properties of profiled 32 antihistamines.

| No. | Compounds | Lipinski Rules | | | | Lipinski's Violations | Bioavailability Score | TPSA ($\text{Å}^2$) |
| | | MW | HBA | HBD | MLog P | | | |
| | | <500 | <10 | ≤5 | ≤4.15 | ≤1 | >0.1 | <140 |
| 1 | Diphenhydramine | 255.35 | 2 | 0 | 3.16 | 0 | 0.55 | 12.47 |
| 2 | Clemastine | 343.89 | 2 | 0 | 4.18 | 1 | 0.55 | 12.47 |
| 3 | Triprolidine | 278.39 | 2 | 0 | 3.16 | 0 | 0.55 | 16.13 |
| 4 | Hydroxyzine | 374.9 | 4 | 1 | 2.45 | 0 | 0.55 | 35.94 |
| 5 | Cyproheptadine | 287.4 | 1 | 0 | 4.46 | 1 | 0.55 | 3.24 |
| 6 | Promethazine | 284.42 | 1 | 0 | 3.84 | 0 | 0.55 | 31.78 |
| 7 | Antazoline | 265.35 | 1 | 1 | 2.89 | 0 | 0.55 | 27.63 |
| 8 | Dimetindene | 292.42 | 2 | 0 | 3.39 | 0 | 0.55 | 16.13 |
| 9 | Ketotifen | 309.43 | 2 | 0 | 3.12 | 0 | 0.55 | 48.55 |
| 10 | Terfenadine | 471.67 | 3 | 2 | 4.80 | 1 | 0.55 | 43.70 |
| 11 | Loratadine | 382.88 | 3 | 0 | 3.72 | 0 | 0.55 | 42.43 |
| 12 | Ebastine | 469.66 | 3 | 0 | 4.73 | 1 | 0.55 | 29.54 |
| 13 | Cetirizine | 388.89 | 5 | 1 | 2.35 | 0 | 0.55 | 53.01 |
| 14 | Rupatadine | 415.96 | 3 | 0 | 4.03 | 0 | 0.55 | 29.02 |
| 15 | Mizolastine | 432.49 | 4 | 1 | 3.59 | 0 | 0.55 | 70.05 |
| 16 | Emedastine | 302.41 | 3 | 0 | 1.91 | 0 | 0.55 | 33.53 |
| 17 | Azelastine | 381.9 | 3 | 0 | 4.28 | 1 | 0.55 | 38.13 |
| 18 | Bilastine | 463.61 | 5 | 1 | 3.29 | 0 | 0.55 | 67.59 |
| 19 | Desloratadine | 310.82 | 2 | 1 | 3.66 | 0 | 0.55 | 24.92 |
| 20 | Fexofenadine | 501.66 | 5 | 3 | 3.86 | 1 | 0.55 | 81.00 |
| 21 | Levocetirizine | 388.89 | 5 | 1 | 2.35 | 0 | 0.55 | 53.01 |
| 22 | Buclizine | 433.03 | 2 | 0 | 5.38 | 1 | 0.55 | 6.48 |
| 23 | Trimeprazine | 298.45 | 1 | 0 | 4.08 | 0 | 0.55 | 31.78 |
| 24 | Carbinoxamine | 290.79 | 3 | 0 | 2.16 | 0 | 0.55 | 25.36 |
| 25 | Tripelenamine | 255.36 | 2 | 0 | 2.32 | 0 | 0.55 | 19.37 |
| 26 | Meclizine | 390.95 | 2 | 0 | 4.78 | 1 | 0.55 | 6.48 |
| 27 | Methdilazine | 296.43 | 1 | 0 | 4.08 | 0 | 0.55 | 31.78 |
| 28 | Dexchlorpheniramine | 274.79 | 2 | 0 | 3.04 | 0 | 0.55 | 16.13 |
| 29 | Dimenhydrinate | 469.96 | 5 | 1 | 2.30 | 0 | 0.55 | 85.15 |
| 30 | Brompheniramine | 319.24 | 2 | 0 | 3.16 | 0 | 0.55 | 16.13 |
| 31 | Chlorpheniramine | 274.79 | 2 | 0 | 3.04 | 0 | 0.55 | 16.13 |
| 32 | Cyclizine | 266.38 | 2 | 0 | 3.01 | 0 | 0.55 | 6.48 |

### 3.2. Targets Related to Antihistamines or COVID-19

A total of 794 targets were connected directly with the number of 32 antihistamines, which were identified via two bioinformatics databases (SEA and STP) (Supplementary Table S1). A total of 198 targets were overlapping between the SEA and STP databases, which was visualized by Venn Diagram plotter (Figure 3A) (Supplementary Table S1). Then, the number of 622 targets that responded directly to COVID-19 infection was identified by PubChem, consequently, a total of 15 targets were overlapping between the overlapping 198 targets and the 622 targets associated with COVID-19 (Figure 3B) (Supplementary Table S2).

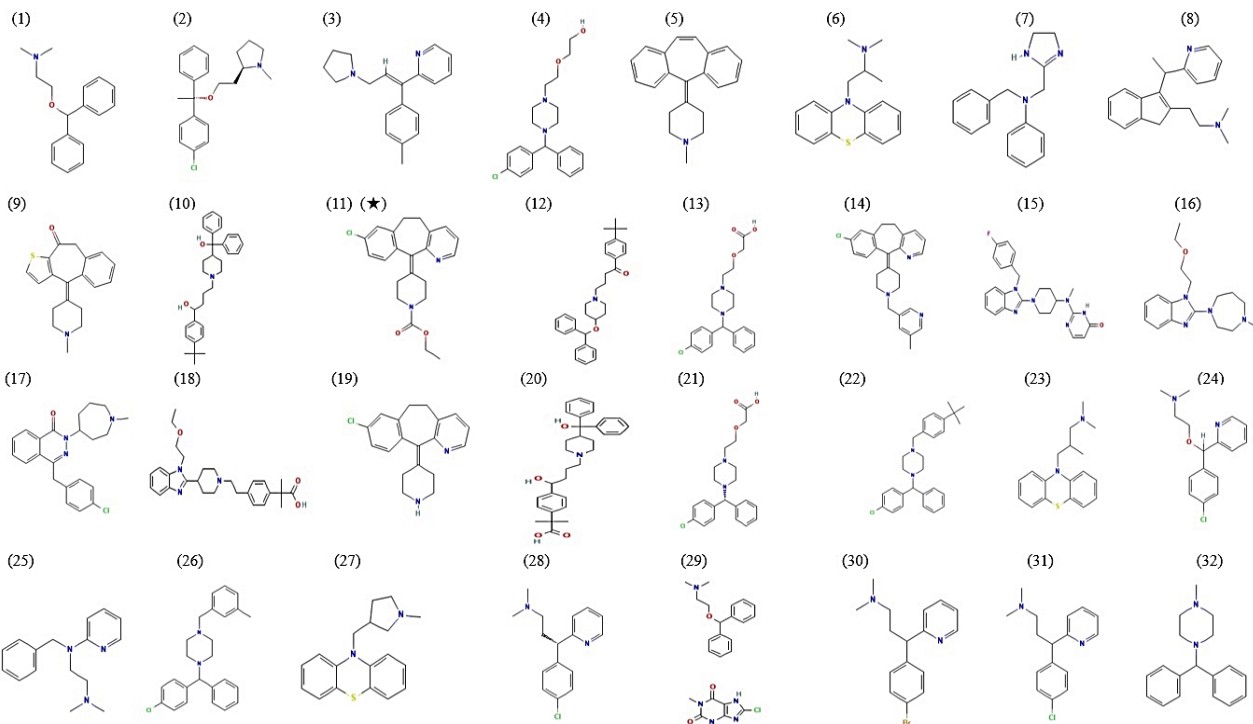

**Figure 2.** Chemical structure of 32 antihistamines. (**1**) Diphenhydramine; (**2**) Clemastine; (**3**) Triprolidine; (**4**) Hydroxyzine; (**5**) Cyproheptadine; (**6**) Promethazine; (**7**) Antazoline; (**8**) Dimetindene; (**9**) Ketotifen; (**10**) Terfenadine; (**11**) (★) Loratidine; (**12**) Ebastine; (**13**) Cetirizine; (**14**) Rupatadine; (**15**) Mizolastine; (**16**) Emedastine; (**17**) Azelastine; (**18**) Bilastine; (**19**) Desloratadine; (**20**) Fexofenadine; (**21**) Levocetirizine; (**22**) Buclizine; (**23**) Trimeprazine; (**24**) Carbinoxamine; (**25**) Tripelenamine; (**26**) Meclizine; (**27**) Methdilazine; (**28**) Dexchlorpheniramine; (**29**) Dimenhydrinate; (**30**) Brompheniramine; (**31**) Chlorpheniramine; (**32**) Cyclizine; (★): A key antihistamine against COVID-19.

### 3.3. Identification of a Key Pharmacological Pathway of Antihistamines on COVID-19

The results of KEGG pathway enrichment analysis suggested that 15 targets were importantly enriched in 7 pathways (*p*-value < 0.05) (Table 2). Based on STRING analysis, the seven pathways were considered as crucial mechanisms that responded to COVID-19 infection. Among the seven pathways, we observed that the neuroactive ligand–receptor interaction (KEGG ID: hsa04080) with the lowest rich factor might be the most critical pharmacological mechanism against COVID-19 (Figure 4).

**Table 2.** Targets in seven pharmacological pathways enrichment associated with COVID-19.

| KEGG ID | Targets | RichFactor | False Discovery Rate |
|---|---|---|---|
| hsa00220:Arginine biosynthesis | NOS2, NOS3 | 0.0952 | 0.0125 |
| hsa04720:Long-term potentiation | GRRIN2B, RPS6KA2, RPS6KA2 | 0.0469 | 0.0026 |
| hsa00330:Arginine and proline metabolism | NOS2, NOS3 | 0.0417 | 0.0497 |
| hsa04931:Insulin resistance | NOS3, RPS6KA2, RPS6KA2 | 0.0374 | 0.0006 |
| hsa04071:Sphingolipid signaling pathway | NOS3, FYN, S1PR5, OPRD1 | 0.0345 | 0.0006 |
| hsa04714:Thermogenesis | PPARG, RPS6KA2, RPS6KA2 | 0.0131 | 0.0497 |
| hsa04080:Neuroactive ligand–receptor interaction | GRIN2B, OPRM1, OPRD1, S1PR5 | 0.0121 | 0.0111 |

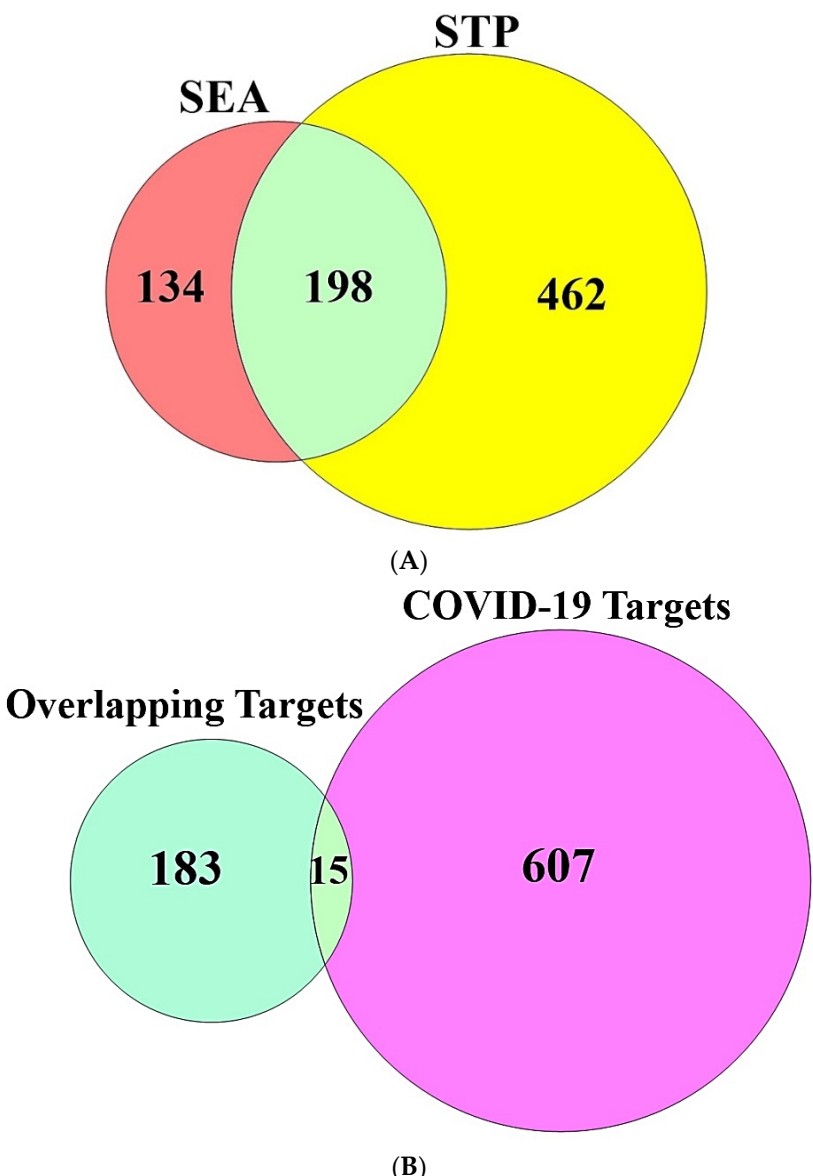

(**A**)

(**B**)

**Figure 3.** (**A**) Overlapping targets (198 targets) of antihistamines associated with targets identified by SEA (332 targets) and STP (660 targets); (**B**) Overlapping targets (15 targets) between the overlapping 198 targets and 622 targets related to COVID-19.

*3.4. Pathways-Targets-Antihistamines (PTA) Network Construction and Analysis*

The 7 pathways—11 targets—30 antihistamines were constructed with 48 nodes and 125 edges (Figure 5). The connectivity lines between the nodes stand for the functional relativeness of these nodes. Among the number of 32 antihistamines, both antazoline and dimenhydrinate have no connectivity with the pathways against COVID-19. Besides, among the number of the final 15 targets, *STGMAR1*, *APP*, *CCR5*, and *NR1I2* have no association with the pathways against COVID-19. The nodes stood for a total number of pathways, targets, and antihistamines. The edges indicated the relationships of the three elements. Among the 11 targets, *NOS3* has the highest degree (4), followed by *RPS6KA2* (3), and *RPS6KA3* (3). Additionally, among the 30 antihistamines, Mizolastine has the highest degree (16), followed by Bilastine (12), Clemastine (11), Fexofenadine (8), Triprolidine (8), Ebastine (8), and Azelastine (8).

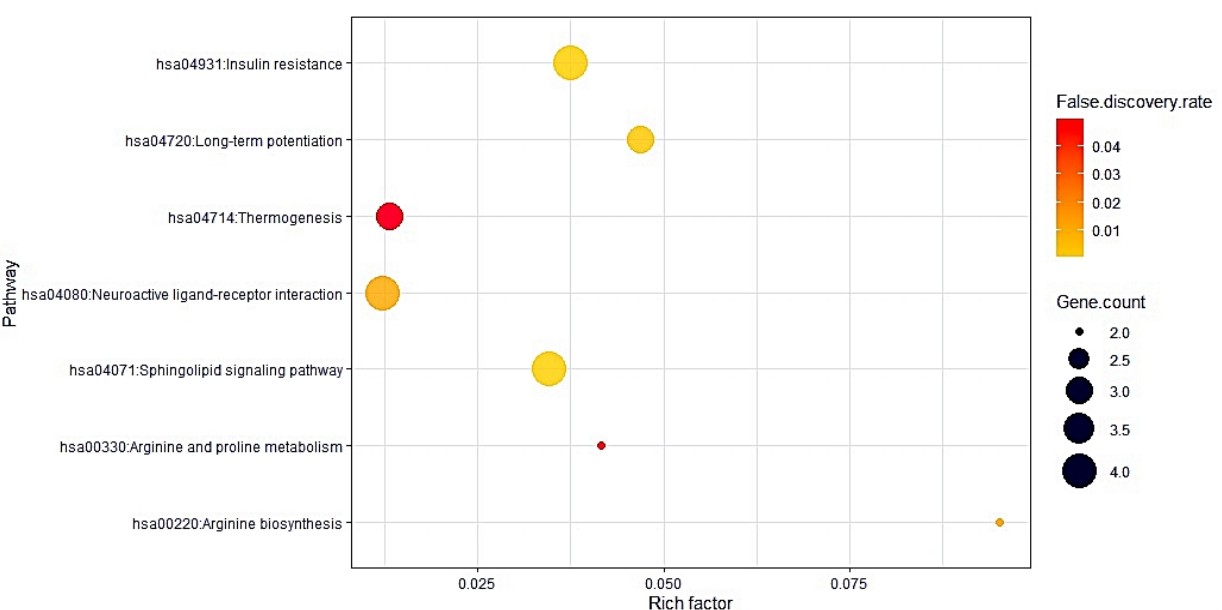

**Figure 4.** A bubble chart of seven potential pharmacological pathways related to the occurrence and development of COVID-19.

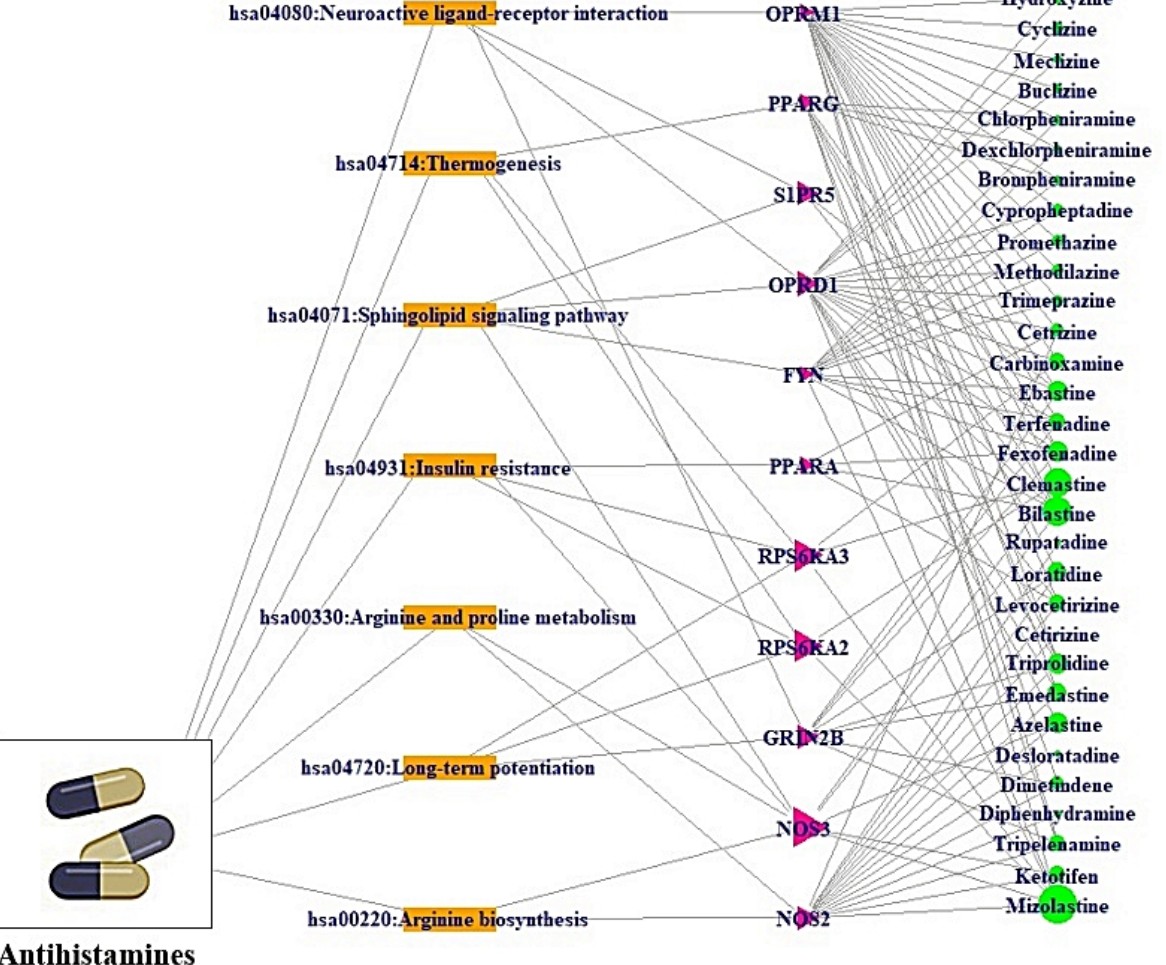

**Figure 5.** Pathways-Targets-Antihistamines (PTA) networks against COVID-19. Orange square: Pathway; Red triangle: Target; Green circle: Antihistamine.

### 3.5. COVID-19 Target PPI Network Analysis

The antihistamines—COVID-19 target PPI network comprised 13 nodes and 19 edges (Figure 6A) (Table 3), and the subnetwork was obtained via sorting out the upper 30%-Degree of Centrality, constituting five nodes and five edges (Figure 6B) (Table 4). After selecting the upper 30%—Betweenness Centrality (BC) of the subnetwork, noticeably, a hub target (GRIN2B) was isolated (Figure 6C). Based on the topological analysis, a sole target, GRIN2B, was considered as a core target in the PPI network analysis.

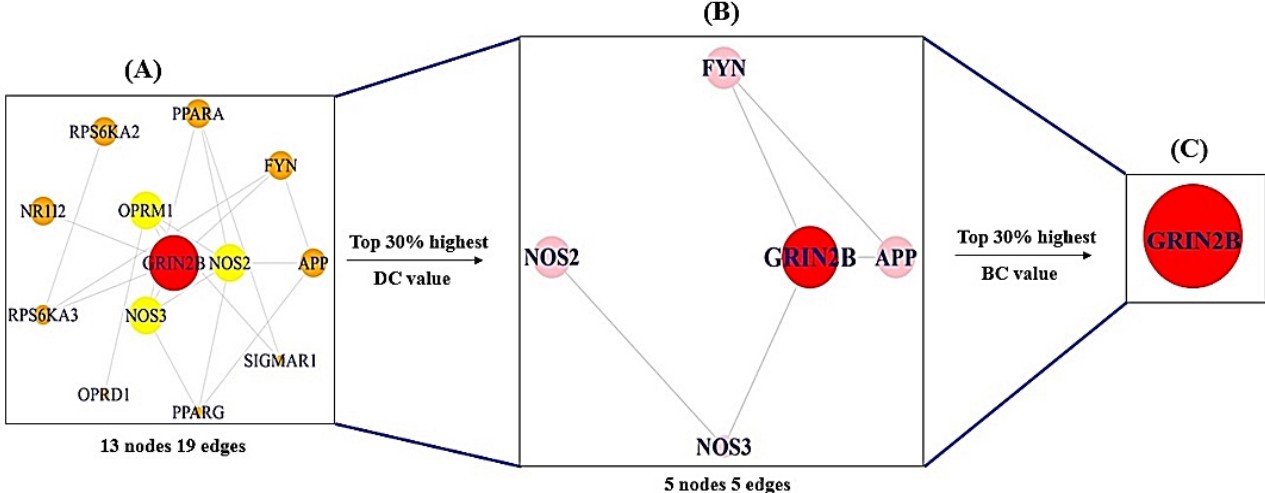

**Figure 6.** Topological analysis of protein–protein interaction networks. (**A**) The PPI network of antihistamines against COVID-19; (**B**) The PPI network of crucial targets extracted from A; (**C**) A final key target for alleviation of COVID-19 extracted from B.

**Table 3.** The values of Degree of Centrality (DC) and Betweenness Centrality (BC) on antihistamines—COVID-19 targets.

| No. | Target | DC | BC |
|-----|--------|----|----|
| 1 | GRIN2B | 4 | 1 |
| 2 | NOS2 | 4 | 0 |
| 3 | APP | 3 | 0 |
| 4 | NOS3 | 2 | 0.43 |
| 5 | FYN | 2 | 0.04 |
| 6 | OPRM1 | 1 | 0.39 |
| 7 | PPARA | 1 | 0.13 |
| 8 | OPRD1 | 1 | 0 |
| 9 | RPS6KA2 | 1 | 0 |
| 10 | PPARG | 0 | 0 |
| 11 | RPS6KA3 | 0 | 0 |
| 12 | NR1I2 | 0 | 0 |
| 13 | SIGMAR1 | 0 | 0 |

**Table 4.** The values of Degree of Centrality (DC) and Betweenness Centrality (BC) of significant targets extracted from Figure 6A.

| No. | Target | DC | BC |
|-----|--------|----|----|
| 1 | GRIN2B | 1 | 1 |
| 2 | APP | 2 | 0 |
| 3 | FYN | 1 | 0 |
| 4 | NOS2 | 1 | 0 |
| 5 | NOS3 | 0 | 0 |

### 3.6. Molecular Docking Test

A total of eight antihistamines (Loratadine, Fexofenadine, Triprolidine, Ebastine, Dimetindene, Terfenadine, Tripelenamine, and Emedastine) and a hub target (GRIN2B) were selected via network pharmacology, with considerable affinity. A molecular docking test (MDT) was performed to verify the binding stability between eight antihistamines and GRIN 2B at the molecular degree. The AutoDockTools-1.5.6 software was utilized for MDT, the docking score is shown in Table 5. The lower docking score (the higher the negative value), the higher the stable conformer is between the ligand and the protein. The MDT results indicated that the docking scores of Loratadine, Fexofenadine, Triprolidine, Ebastine, Dimetindene, and Terfenadine to the GRIN2B target were $<-6.0$ kcal/mol, suggesting that these six antihistamines can exert a stable binding effect with the GRIN2B target. Two other antihistamines (Tripelenamine, and Emedastine) had invalid binding scores ($>-6.0$ kcal/mol). The threshold of AutoDockTools-1.5.6 software is $<-6.0$ kcal/mol, which can represent active ligand on target [26].

**Table 5.** The docking score and interactions of a key antihistamine (Loratadine) binding to a key target (GRIN2B).

| Protein | Ligand | PubChem ID | Binding Energy (kcal/mol) | Grid Box | | Hydrogen Bond Interactions | Hydrophobic Interactions |
|---|---|---|---|---|---|---|---|
| | | | | Center | Dimension | Amino Acid Residue | Amino Acid Residue |
| GRIN 2B (PDB ID: 7EU8) | Loratidine | 3957 | −7.3 | x = 128.688 | size_x = 40 | N/A | Trp391, Glu163, Tyr164 |
| | | | | y = 128.088 z = 133.365 | size_y = 40 size_z = 40 | | Asp165, Val390, Tyr389 Asp477, Tyr476, His405 Thr475, Trp166 |
| | Fexofenadine | 3348 | −6.8 | x = 128.688 | size_x = 40 | Trp166, Trp391, Asp165 | Ser469, Phe474, Thr475 |
| | | | | y = 128.088 z = 133.365 | size_y = 40 size_z = 40 | | Tyr476, Val390, Pro435 Tyr164 |
| | Triprolidine | 5282443 | −6.7 | x = 128.688 y = 128.088 z = 133.365 | size_x = 40 size_y = 40 size_z = 40 | Arg755 | Ala734, Glu531, Leu797 Met789, Phe460, Phe529 Ile530, Glu793, Leu752 |
| | Ebastine | 3191 | −6.6 | x = 128.688 y = 128.088 z = 133.365 | size_x = 40 size_y = 40 size_z = 40 | Ile190 | Ile691, Gln487, Glu522 Trp498, Asn521, Tyr526 Glu191, Gly196 |
| | Dimetindene | 21855 | −6.5 | x = 128.688 y = 128.088 z = 133.365 | size_x = 40 size_y = 40 size_z = 40 | N/A | Leu792, Leu795, Asp463 Asn432, Trp796, Lys458 Thr701, Arg673, Leu699 Glu698 |
| | Terfenadine | 5405 | −6.4 | x = 128.688 y = 128.088 z = 133.365 | size_x = 40 size_y = 40 size_z = 40 | N/A | Phe474, Trp166, Val390 Trp391, Pro435, Tyr164 Asp165, Thr433, Thr475 Tyr476 |
| | Tripelenamine | 5587 | −5.6 | x = 128.688 y = 128.088 z = 133.365 | size_x = 40 size_y = 40 size_z = 40 | N/A | Leu752, Glu793, Phe529 Ile530, Leu797, Asn737 Ala734, Arg755, Ala794 Glu790 |
| | Emedastine | 3219 | −5.5 | x = 128.688 y = 128.088 z = 133.365 | size_x = 40 size_y = 40 size_z = 40 | Trp166 | Ser469, Pro435, Phe474 Tyr164, Val434, Thr433 Tyr476 |

Additionally, Loratadine, the most significant antihistamine, was the key antihistamine as a COVID-19 alleviator. According to MDT, Loratadine exhibited the best binding scores ($-7.3$ kcal/mol) to the GRIN2B target, which is displayed in Figure 7.

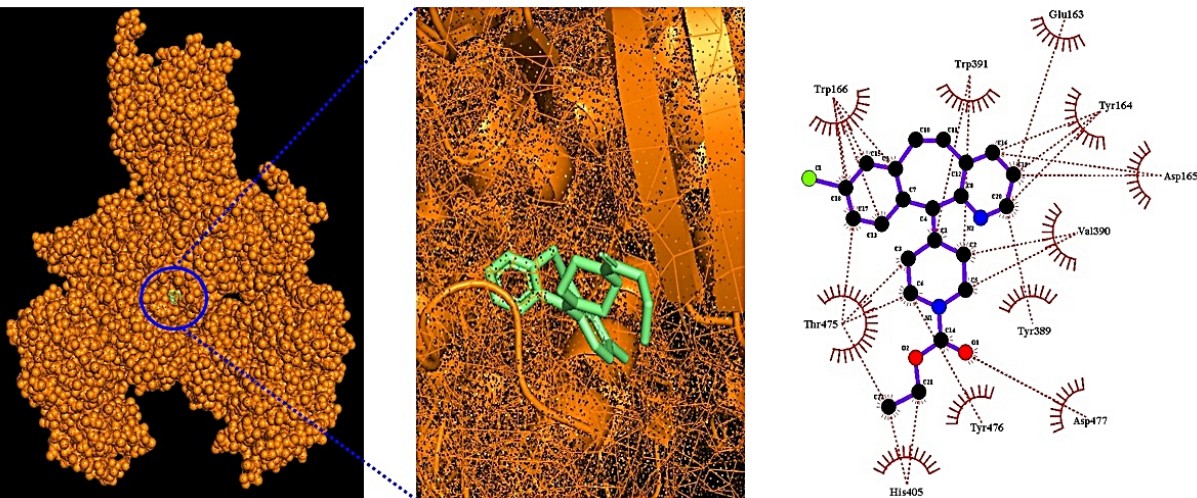

**Figure 7.** Molecular docking test between Loratadine and GRIN2B (PDB ID: 7EU8). The Loratadine structure was displayed as a green stick, the target structure was presented as an orange color and the interacted residues were exhibited as a dotted red line.

## 4. Discussion

Antihistamines are important agents to inhibit inflammatory responses: allergic rhinitis [27], asthma [28], anaphylaxis [29], and pneumonia [30]. Most recently, COVID-19 patients treated with some antihistamines, including Loratadine, have improved notably within 48 h [31]. However, studies have not yet been reported on the underlying pharmacological mechanism(s), ultimate antihistamine(s), and target(s) against COVID-19.

In this study, we investigated the pathways-targets-antihistamines (PTA) network and observed that Loratadine, Fexofenadine, Triprolidine, Ebastine, Dimetindene, and Terfenadine had a valid correlation on GRIN2B target on the neuroactive ligand–receptor interaction (KEGG ID: hsa04080), representing significant alleviative effects for COVID-19. A study demonstrated that the induction of GRIN2B is subsequent to neurotoxicity in the brain, due to the neuro-inflammation pathway [32]. It can be speculated that GRIN2B is a critical target to diminish the inflammatory responses. In fact, nodes with high Betweenness Centrality (the measurement of the shortest path between other nodes) are of significant component to be related to key multiple therapeutic uses and thus a notable potential value to be repurposed [33]. It implies that GRIN2B with the highest Betweenness Centrality might be the most important target of antihistamines in COVID-19. The Antihistamines–targets network represented that the therapeutic efficacy of antihistamines on COVID-19 was directly related to 13 targets. The results of KEGG pathway enrichment analysis of 13 targets indicated that 7 pharmacological pathways were directly associated with the development of COVID-19, suggesting that these pathways might be the mechanisms of antihistamines against COVID-19. The associations with the seven pathways with COVID-19 were briefly discussed as follows. Insulin resistance: The subjects with diabetes are more vulnerable to COVID-19 infection than healthy subjects, also, COVID-19 accelerates insulin resistance in the subjects, and leads to chronic metabolic disorders that had not occurred before infection [34]. Long-term potentiation is defined as a processing that keeps incessant strengthening of synapses that induces to a persistent increase in synaptic plasticity between neurons [35]. A study demonstrated that ACE knockout mice lead to the impairment of cognitive function in brain due to the dysfunction of synaptic plasticity [36]. It suggests that SARS-CoV-2 could be worsening via the binding to ACE2 on neurons and glial cells, that might act as a target and be susceptible to infection [37]. Thermogenesis: The brown adipose tissue (BAT) plays a critical role to control the heat generation, in particular, in virus side, thermogenesis might be a significant pathway to enhance against viral infection [38,39]. Neuroactive ligand–receptor interaction: Most recently, neuroactive ligand–receptor interaction in MERS-CoV-infection has been upreg-

ulated on Gene Set Enrichment Analysis (GSEA). Thus, it has been speculated that the intervention of neuroactive ligand–receptor interaction might be a key target to ameliorate COVID-19. Sphingolipid signaling pathway: SARS-CoV-2 infection led to an increase in sphingolipid levels in both cells and sera of mice [40]. It implies that sphingolipids might be an important agent to replicate SARS-CoV-2. Arginine and proline metabolism: Arginine has an effect on the accumulation of succinic acid in severe COVID-19 disease, and proline elevates with Interleukin 17A (IL17A) or Interleukin 17 Receptor A (IL17RA) in SARS-CoV-2 infection [41]. Arginine biosynthesis: Arginine administered orally to severe COVID-19 patients has great efficacy in decreasing the duration of hospitalization and lessens the usage of respiratory equipment at 10 days but not at 20 days after treatment [42]. A report also suggested that diphenhydramine, hydroxyzine, and azelastine have potent antiviral efficacy against COVID-19 in vitro [43]. Another study reported that combinational drugs with antihistamines and azithromycin helped ameliorate symptoms of COVID-19 without any noticeable adverse effects in elderly patients [9].

From our network pharmacology analysis based on rich factor in bubble chart, neuroactive ligand–receptor interaction was the lowest rich factor in seven pharmacological pathways, suggesting that the pathway might be a key mechanism against COVID-19. Furthermore, GRIN2B with the highest degree of value in PPI network, with the highest Betweenness Centrality (BC) in topological analysis, and correlated with neuroactive ligand–receptor interaction was considered as a core target of antihistamines in COVID-19. Then, the best antihistamine on GRIN2B was Loratadine, which can conform stably with valid affinity (−7.3 kcal/mol). Therefore, the hub mechanism of antihistamines against COVID-19 might be to inhibit GRIN2B on the neuroactive ligand–receptor interaction by Loratadine as an antagonist.

## 5. Conclusions

To sum up, this study demonstrated that the most potential antihistamine mechanism in alleviating COVID-19 is based on network pharmacology. We suggested that the six antihistamines (Loratadine, Fexofenadine, Triprolidine, Ebastine, Dimetindene, and Terfenadine) played a critical role in COVID-19 by affecting the GRIN2B target as well as inhibiting neuroactive ligand–receptor interaction (KEGG ID: hsa04080). Besides, MDT also showed that Loratadine was the best affinity on the GRIN2B target on neuroactive ligand–receptor interaction (KEGG ID: hsa04080), providing an important basis for further clinical trials.

**Supplementary Materials:** The following supporting information can be downloaded at: https://www.mdpi.com/article/10.3390/cimb44040109/s1, Table S1: The number of 332 targets from SEA, 660 targets from STP, a total of 794 targets from SEA and STP, and the number of overlapping 198 targets between SEA and STP; Table S2: The number of 622 targets related to COVID-19, the number of 15 final targets.

**Author Contributions:** Conceptualization, Methodology, Formal analysis, Investigation, Visualization, Data Curation, Writing—Original Draft, K.-K.O.; Software, Investigation, Data Curation, K.-K.O. and M.A.; Validation, Writing—Review and Editing, M.A.; Supervision, Project administration, D.-H.C. All authors have read and agreed to the published version of the manuscript.

**Funding:** This research did not receive any specific grant from funding agencies in the public, commercial, or not-for-profit sectors.

**Institutional Review Board Statement:** Not applicable.

**Informed Consent Statement:** Not applicable.

**Data Availability Statement:** All data generated or analyzed during this study are included in this published article (and its Supplementary Information Files).

**Acknowledgments:** This study has been worked with the support of a research grant of Kangwon National University in 2022.

**Conflicts of Interest:** The authors declare no conflict of interest.

**Abbreviations**

| | |
|---|---|
| COVID-19 | Coronavirus disease 2019 |
| BC | Betweenness Centrality |
| DC | Degree Centrality |
| GSEA | Gene Set Enrichment Analysis |
| IL17A | Interleukin 17A |
| IL17RA | Interleukin 17 Receptor A |
| KEGG | Kyoto Encyclopedia of Genes and Genomes |
| MDT | Molecular Docking Test |
| PPI | Protein-Protein Interaction |
| PTA | Pathways-Targets-Antihistamines |
| SEA | Similarity Ensemble Approach |
| STP | SwissTargetPrediction |
| TPSA | Topological Polar Surface Area |
| WHO | World Health Organization |

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
