# Peer review of "Network Pharmacology Study to Elucidate the Key Targets of Underlying Antihistamines against COVID-19"

_cimb, doi:10.3390/cimb44040109_

Round 1

Reviewer 1 Report

This is an interesting paper exploring in silico approach to identify new pathway of phamacological interest to fight viral infection.  This is a very promising approach that deserves publication in the present form. 

Author Response

Point 1: This is an interesting paper exploring in silico approach to identify new pathway of phamacological interest to fight viral infection.  This is a very promising approach that deserves publication in the present form.  

Response 1: Thanks for the great your assessment.

Reviewer 2 Report

Dear, Authors
It is very interesting and novel idea which gives a crucial hint about drug repurposing, in particular, antihistamines against COVID-19. 
The analytical concept is based on network pharmacology including in silico.
I think that this concept is applicable to other diseases. 
This manuscript is written well on scientific evidence. 
In this stage, I recommend acceptance without any specific flaws. 

Thanks
Sincerely yours

Author Response

Point 1: Dear, Authors
It is very interesting and novel idea which gives a crucial hint about drug repurposing, in particular, antihistamines against COVID-19. 
The analytical concept is based on network pharmacology including in silico.
I think that this concept is applicable to other diseases. 
This manuscript is written well on scientific evidence. 
In this stage, I recommend acceptance without any specific flaws. 

Thanks
Sincerely yours

Response 1: Thanks for the great your assessment.

Reviewer 3 Report

The article is interesting and is certainly well structured, however small but substantial changes are needed.
The captions of figures 4, 5, and 6 need to be better described. Furthermore, figure 5 is unclear, there is a lot of confusion between the lines and the true meaning of the figure is lost, which is why it must be completely changed.
Discussion: Authors should broaden the discussion by comparing their results with what is already present in the literature, such as:
Reznikov, L.R., Norris, M.H., Vashisht, R., Bluhm, A.P., Li, D., Liao, Y.S.J., Brown, A., Butte, A.J. and Ostrov, D.A., 2021. Identification of antiviral antihistamines for COVID-19 repurposing. Biochemical and biophysical research communications, 538, pp. 173-179.

Morán Blanco, JI, Alvarenga Bonilla, JA, Homma, S., Suzuki, K., Fremont-Smith, P. and Villar Gómez de Las Heras, K., 2021. Antihistamines and azithromycin as a treatment for COVID-19 on primary health care-A retrospective observational study in elderly patients. Pulm Pharmacol Ther, pp. 101989-101989.

Minor revisions:
Line 42: The following references must be added:
10.3390 / healthcare10020319
Line 246: "Long-term potentiation:" can be canceled.

Author Response

The article is interesting and is certainly well structured, however small but substantial changes are needed.
Point 1: The captions of figures 4, 5, and 6 need to be better described. Furthermore, figure 5 is unclear, there is a lot of confusion between the lines and the true meaning of the figure is lost, which is why it must be completely changed.

Response 1: Thanks for the great suggestion. We revised the captions more detail and added the meaning of connectivity line in 3.4 section with yellow highlight.  

Point 2: Discussion: Authors should broaden the discussion by comparing their results with what is already present in the literature, such as:

Reznikov, L.R., Norris, M.H., Vashisht, R., Bluhm, A.P., Li, D., Liao, Y.S.J., Brown, A., Butte, A.J. and Ostrov, D.A., 2021. Identification of antiviral antihistamines for COVID-19 repurposing. Biochemical and biophysical research communications, 538, pp. 173-179.

Morán Blanco, JI, Alvarenga Bonilla, JA, Homma, S., Suzuki, K., Fremont-Smith, P. and Villar Gómez de Las Heras, K., 2021. Antihistamines and azithromycin as a treatment for COVID-19 on primary health care-A retrospective observational study in elderly patients. Pulm Pharmacol Ther, pp. 101989-101989.

Response 2: Thanks for the great suggestion. We added up your suggested good references in Discussion section with highlighted yellow color.

Minor revisions:
Point 3: Line 42: The following references must be added:
10.3390 / healthcare10020319

Response 3: Thanks for the great suggestion. We added up your suggested good reference in the line.

Point 4: Line 246: "Long-term potentiation:" can be canceled.

Response 4: Thanks for the great suggestion. We removed ONE of TWO. Thanks a lot.

Round 2

Reviewer 3 Report

The authors have improved the paper although Figure 5 should be clearer, the lines remain confusing